cognition/psychology

facial emotion, affective processing, genetic algorithm, individual differences

**Author for correspondence:**
Christina O. Carlisi
e-mail: c.carlisi@ucl.ac.uk

†Authors contributed equally to this manuscript.

# Using genetic algorithms to uncover individual differences in how humans represent facial emotion

Christina O. Carlisi[1], Kyle Reed[2], Fleur G. L. Helmink[3], Robert Lachlan[4], Darren P. Cosker[2], Essi Viding[1,†] and Isabelle Mareschal[5,†]

[1]Division of Psychology and Language Sciences, Developmental Risk and Resilience Unit, University College London, 26 Bedford Way, London WC1H 0AP, UK
[2]Department of Computer Science, University of Bath, 1 West, Claverton Down, Bath BA2 7AY, UK
[3]Erasmus University Medical Center, s-Gravendijkwal 230, Rotterdam 3015 CE, The Netherlands
[4]Department of Psychology, Royal Holloway University of London, Wolfson Building, Egham TW20 0EX, UK
[5]School of Biological and Chemical Sciences, Department of Psychology, Queen Mary University of London, G. E. Fogg Building, Mile End Road, London E1 4DQ, UK

COC, 0000-0002-0942-8586

Emotional facial expressions critically impact social interactions and cognition. However, emotion research to date has generally relied on the assumption that people represent categorical emotions in the same way, using standardized stimulus sets and overlooking important individual differences. To resolve this problem, we developed and tested a task using *genetic algorithms* to derive assumption-free, participant-generated emotional expressions. One hundred and five participants generated a subjective representation of happy, angry, fearful and sad faces. Population-level consistency was observed for happy faces, but fearful and sad faces showed a high degree of variability. High test–retest reliability was observed across all emotions. A separate group of 108 individuals accurately identified happy and angry faces from the first study, while fearful and sad faces were commonly misidentified. These findings are an important first step towards understanding individual differences in emotion representation, with the potential to reconceptualize the way we study atypical emotion processing in future research.

## 1. Introduction

Facial expressions are ubiquitous in our everyday environments, conveying information about emotional states and intentions.

How humans represent and interpret facial expressions shape relationships and guide social interactions [1,2], but it has been debated whether basic emotional expressions are universally recognized [3–6]. However, much of the research on emotion representation has relied upon well-validated yet over-generalized stimulus sets of emotional expressions, potentially limiting detection of nuanced features underpinning emotion recognition at the individual level (i.e. individual differences). A more precise measure of such features would represent a step-change in how we conduct and interpret emotion research.

Emotion perception involves processing information communicated by morphological facial changes, e.g. eyes or mouth opening, frowning or smiling [7]. These morphological changes convey the expresser's internal feelings or sentiment [8]. Emotion recognition is thus a calculated process by which perceptual and consequent affective information must be interpreted using acquired semantic knowledge and contextual information that varies widely among individuals [9,10]. However, the factors underlying these individual differences are unclear. Differences may reflect a lack of sensitivity to emotional expressions (i.e. a face looks sad to an individual, but their response is not typical for a sad face), a different internalized representation of the emotion (i.e. what an individual thinks 'sad' should look like when represented on a face) or individual differences in low-level visual processing which compromises the perception of certain expressions more than others. Distinguishing between these different mechanisms of emotion processing and how they vary among individuals across a population is important; at the extreme end, such individual differences can result in failures to appropriately interpret emotional expressions and can increase one's risk of psychopathology [11,12].

Methods for testing emotion recognition typically rely on the assumption that emotions are discrete categories, which requires broad agreement across observers. However, it is likely that emotional expressions are represented dimensionally within these categories [8], varying subtly from person to person. Many experiments are limited in their ability to index this variation within categories because they use pre-established sets of facial stimuli designed to universally represent basic emotions [13,14], when in reality, emotion categories are not represented in the same way by all individuals. Additionally, studies of emotion processing that use a forced-choice response to identify a given facial stimulus may artificially constrain the detection of individual differences [5]. Therefore, the present study sought to understand this fine-grained variation in emotion representation at the individual level within traditional emotion categories through the indexing of participant-generated emotional facial expressions.

Participant-generated facial expressions have been of interest to the computer animation community for some time [15]. Since the introduction of performance-driven animation [16], there have been numerous implementations of data-driven methods, including principle component analysis (PCA) [17,18], Gaussian process latent variable models (GPLVMs) [19,20] and autoencoders [21,22] to create realistic, performance (i.e. participant) driven graphics [23,24]. However, these techniques often rely on the secondary judgement of a skilled artist and are limited in their ability to allow users to explore and interpret resulting graphics. More recently, machine learning techniques such as generative adversarial networks (GANs) [25,26] and verbal crowd-shaping [27] have been applied to create computer-generated facial expressions and body images based on sampled parameters and descriptive labels from a population. While promising, these methods rely on unsupervised learning and classification, requiring vast training datasets and fully trained networks for optimal performance. Therefore, such approaches may not be best-suited to index the nuanced and subjective nature of faces and emotion representation. New implementations of existing methods such as the genetic algorithm (GA)—thus termed because of its procedural similarities to evolutionary processes—may solve the longstanding problem of optimally yet efficiently capturing an individual's internal representation of facial emotion. GAs efficiently explore a parameter space that exceeds the limits tested using standard face stimuli sets and are particularly well-suited to multi-dimensional problems such as efficiently searching 'face-space', where a number of different features (e.g. eye shape, mouth curvature) contribute to an overall facial expression [28,29]. GAs have been widely and successfully used in ecology and forensic settings, where they enable eyewitnesses to generate facial composites with greater accuracy compared with traditional sketches [30,31]. However, GAs have not been widely applied to facial emotion research. Prior studies have predominantly focused on predetermined emotional stimuli recorded from actors which do not quantify the participant's own perceptual representation of an emotional expression. While some studies have used data-driven approaches with non-static stimuli [2,24,32–34], these methods are time-consuming, usually requiring a large number of trials, and therefore ill-suited for large-scale testing (or testing child or clinical populations). Moreover, a true depiction of an individual's internal representation of facial expressions has traditionally relied on highly skilled artists, as is frequently the case in computer animation. The GA

presents a new heuristic that allows the generation of individuals' internal representations of expressions without the lengthy process of artist renderings.

To address the methodological gaps outlined above, our study developed a cognitive task applying a GA to the individualized generation of facial emotional expressions. The task allowed participants to quickly and efficiently create facial expressions based on their own ideals and internal representation of a given emotion. Across a number of trials, participants were free to evolve their ideal emotional expressions according to their own perception, as opposed to having these ideals imposed by experimenter decisions according to pre-selected, standardized emotional stimuli. This study had two primary objectives, (i) to present for the first time the use of a GA in facilitating participant-driven generation of facial emotional expressions, and (ii) to use this task to examine individual differences in the way participants represent and perceive these expressions. We applied this GA framework in which participants' choices directly facilitated stimulus creation, to quantify individual differences in emotion representation not typically captured by traditional paradigms where stimuli are pre-selected as part of the task design and are thus subject to experimenter-imposed bias. This approach opens up new possibilities for the study of human emotion perception, social cognition and affective processing.

# 2. Methods

## 2.1. Overview of the genetic algorithm

The GA facilitated participant-driven generation of individualized static facial expressions based on one's internal representation of happy, angry, sad and fear faces. GAs adapt a reinforcement learning approach to produce individualized images based on user feedback. Analogous to evolution, GAs rely on sampling and evaluation of parameter combinations to assess the fitness of outcomes (i.e. expressions are evolved over several *generations* of *sampling*, *selection*, *breeding* and *mutation* until the *gene pool* converges or *evolves* toward the participant's ideal expression). This scheme is 'user-guided', i.e. the participant performs the sampling/evaluation of solutions to produce individualized expressions.

A comprehensive description of the GA is presented in the electronic supplementary material, eAppendix-1 and in [35]. Expressions were generated from an animator's face rig [36] (https://www. camera.ac.uk/) in [37] (https://www.autodesk.co.uk/) by combining bound weights between 0 and 1 from 42 facial features (defined as '*blendshapes*', e.g. lip curl, jaw openness) governed by the Facial Action Coding System (FACS) [7]. Blendshapes were chosen based on an animator's (K.R.) expert opinion of the most significant shapes from a generic facial rig to cover a sufficient range of expression. The rig was adapted from a larger rig where blendshape correctives were removed and left/right blendshapes were combined into a single dual-sided action. This is necessary because without such constraints, correctives and asymmetry increase the likelihood of implausible, non-realistic expressions being created. To apply corrective shapes to improve realism of the faces generated, we sampled from rig controls, which incorporates correctives. This procedure is described at length in [35]. Obtaining a suitable combination of blendshapes in the faces presented to participants also necessitates incorporating an appropriate mutation rate in the GA, described in terms of the difference in blendshape weights bounded [0, 1]. To achieve this, a random value was assigned between 0.25 and 0.75 [35] (for complete details, see electronic supplementary material, eAppendix-1).

All participants were shown computer-generated faces modelled on the same artist-rendered facial rig. Participants selected a subset of expressions they determined most closely represented a specified emotion, and the process repeated over a number of iterations, with each iteration generating a new set of facial expressions based on features selected for in the previous subset (with noise added, or random mutations) (figure 1). Participants completed six iterations of the GA, selecting 3 out of 10 faces on each iteration, with participants' selections driving the feature selection scheme, as participants choose sample faces with the most desired characteristics for their search. The combination of six generations and 10 faces per generation was chosen to strike a balance between avoiding information overload for participants on each trial, increasing efficiency, presenting an adequate number of generations [38], minimizing participant 'burn-out' (i.e. boredom) since the faces become more similar with each iteration, and keeping the faces easy to view on a single screen [35]. To compensate for this relatively small population of test faces, we employed a *coarse-to-fine* approach to more extensively cover the search space. This process involves choosing facial features with the greatest variability for random sampling and mutation in initial generations, with more nuanced

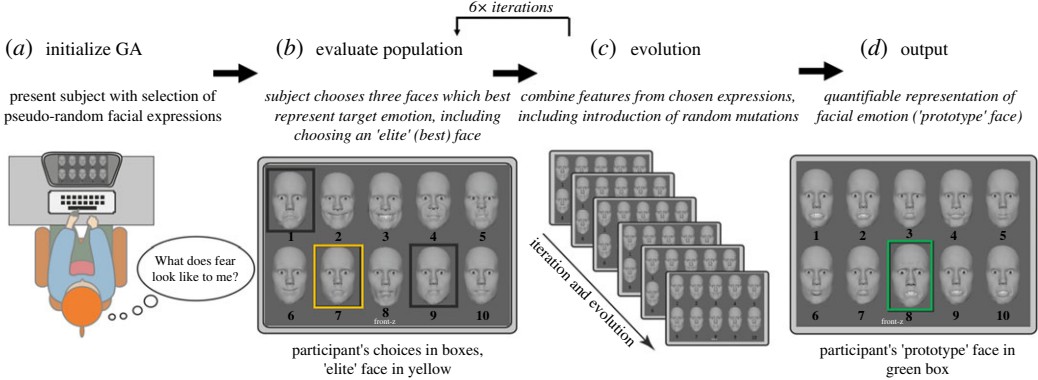

**Figure 1.** Schematic overview of task procedure. (*a*) Step one of the genetic algorithm procedure. The participant is presented with 10 faces from a pre-selected population and a target emotion (e.g. 'fear') to identify. They are asked to reflect on how they subjectively represent the target emotion. (*b*) The participant relies on this representation to guide selection of three faces from the population which are most aligned with the target emotion, including an 'elite' face that is the best representation (in yellow). (*c*) This process is repeated and the population is evolved. (*d*) After a fixed number of generations, a final 'prototype' facial expression is generated that approximates that participant's internalized representation of the target emotion (in green). This is the expression that is then quantified and analysed.

changes in expressions chosen in later generations (for extended technical details of the coarse-to-fine approach and overall GA evolution procedure, see electronic supplementary material, eAppendix-1).

## 2.2. Participants

A total of 105 adults between 18 and 61 years of age (29% male, mean age 27 ± 8 years, 51% Caucasian, 42% Asian, 2% Black, 5% other) were recruited via a community convenience sample and completed the GA task on one occasion. An *a priori* power calculation revealed that in order to detect medium effect sizes (Cohen's $f = 0.25$) with 95% power, a minimum sample size of $N = 55$ was required. Therefore, anticipating 10% drop-off, 60 participants from the original convenience sample repeated the task on two more occasions, once approximately 30 min later, and again after 30–45 days. All participants gave written informed consent, and the study was approved by the UCL Research Ethics Committee (Z6364106/2013/07/37). Data were processed in accordance with EU General Data Protection Regulations.

## 2.3. Behavioural task

For each generation, participants selected 3 out of 10 faces that they thought best represented the target emotion (happy, angry, sad, fearful). Participants were then instructed to select one of their chosen faces which they thought best represented the target emotion (the 'elite' face). This process was repeated six times. On a given trial, participants were free to amend their selections and were not constrained by explicit time limits (though they were instructed to select as quickly yet accurately as possible). Once satisfied with their choices, participants evolved the generation. On the subsequent generation, participants were presented with a new set of 10 faces evolved from the previous generation, and they repeated the selection and evolution process. After six generations, blendshape weights for the final elite face were saved for further analysis. This face was termed the 'prototype face' for that participant, i.e. the participant's optimal representation of the target emotion. The four emotions were tested in separate blocks, the order of which was randomized across participants.

The GA was run on a 33.8 cm Dell laptop (1366 × 768, 60 Hz) at UCL using the Autodesk Maya interface. Participants were seated approximately 57 cm from the screen (1 pixel subtended 1.3 arcmin). Faces were displayed in greyscale and subtended 1.9 × 3.0°.

## 2.4. Statistical analysis

### 2.4.1. Emotion specificity and response variability

Statistical analyses were performed in Matlab [39] (Mathworks, Inc., Natick, MA) and R v. 3.5.0 [40]. Multivariate dispersion analysis [41] assessed variability of 'community structure', i.e. variability of individual

prototype faces from the population norm within each emotion category; the spatial median (i.e. population norm) was computed within 42-dimensional 'face-space' [29] corresponding with the 42 blendshapes, and the average distance (dissimilarity) between each individual data point and its category's spatial median—defined in the principal coordinate of face-space—was calculated using Euclidian distances from this median as the dissimilarity measure. Levene's F-statistic was calculated to compare the average distance of individual data points to the population norm for each emotion category, and the significance of dissimilarity ($p$-value) was calculated by permuting least-absolute-deviation residuals and recalculating the F-statistic.

The t-distributed stochastic neighbour embedding (t-SNE) enabled dimensionality reduction of the 42-dimensional face-space and visual evaluation of the two-dimensional clustering of faces based on their respective categories (happy, angry, sad, fear) (for complete details, see electronic supplementary material, eAppendix-2).

To assess the most important features in forming prototype expressions for each emotion, each blendshape weight was scaled based on its geometric error from neutral. This resulted in a scaled metric of influence based on the extent to which each blendshape affects the overall mesh geometry. This enabled the ranking of blendshape influence on the overall expression by their scaled degree of change in the mesh.

### 2.4.2. Test–retest reliability and within-subject response variability

To assess reliability of the GA task, we examined whether between-subject individual differences in prototype faces were greater or smaller than within-subject variance across time. This also allowed us to confirm that after six generations, the prototype face was not a result of either a specific mutation or local minimum in the algorithm's iterative process. We first calculated a dissimilarity matrix for each emotion from the Euclidean distances in 42-dimensional face-space between all the faces produced at the three time points. We then coded an individual-identity matrix for each emotion, where each row and column were different faces, and within-subject comparisons were assigned a value of 0 and between-subject comparisons were assigned a value of 1. Mantel tests were used to test for correlations between the individual-identity matrices and the dissimilarity matrices using the 'vegan' package in R [42] with 9999 permutations and using a Spearman test statistic. Resulting distance values from multivariate dispersion analysis within each emotion category (§2.4.1) for each time point were also used to compute two-way mixed-effects intraclass correlation coefficients (ICCs).

## 2.5. External validity

To assess external validity of the prototype faces generated from the GA procedure, 26 faces (25% of the total number of prototypes generated from time point 1) were randomly selected for each emotion, resulting in a total of 104 stimuli. We used Amazon's Mechanical Turk (https://www.mturk.com/) to recruit 108 individuals (66% male; 67% white, 30% Asian, 7% black/mixed) between ages 18 and 65 ($M = 35$ years). Faces were presented using an online platform designed for behavioural experiments (www.gorilla.sc), and participants described, using one word in a free-text response box, the emotion that they perceived from the face.

Responses were coded independently and cross-validated by C.O.C. and F.G.L.H. as either (i) an exact match (happy/happiness, angry/anger, sad/sadness, fear/fearful), (ii) correct (synonymous with happy/angry/sad/fearful as defined by Merriam-Webster online thesaurus (www.merriam-webster.com/thesaurus) (iii) incorrect (a plausible description of a different emotion, but not a synonymous match for the target emotion) or (iv) anomalous (indecipherable, e.g. 'd' or '2'; not consistent with a standard descriptor of basic facial emotion, e.g. 'flirty', 'lying'; or ambiguous, e.g. 'intense', 'indifferent').

# 3. Results

## 3.1. Emotion specificity and response variability

Figure 2$a$ shows distributions of Euclidean distances for prototype faces from the spatial median (conceptualized as the population 'norm'). Population-level deviation from the norm was lowest for happy, with the highest density of distances closest to 0, and greatest for fear, with the highest density of distances furthest from 0. This was confirmed by comparisons of distances from the spatial median between each emotion: participants' prototype happy faces deviated less from the population norm compared with sad [$F_{1,358} = 98.79$, $p < 0.0001$, adjusted $R^2 = 0.21$], angry [$F_{1,358} = 134.10$, $p < 0.0001$, adjusted $R^2 = 0.27$] and fear [$F_{1,358} = 14.67$, $p < 0.001$, adjusted $R^2 = 0.04$]. Fearful faces deviated more from the spatial median compared with sad [$F_{1,358} = 29.12$, $p < 0.0001$, adjusted $R^2 = 0.07$], and angry

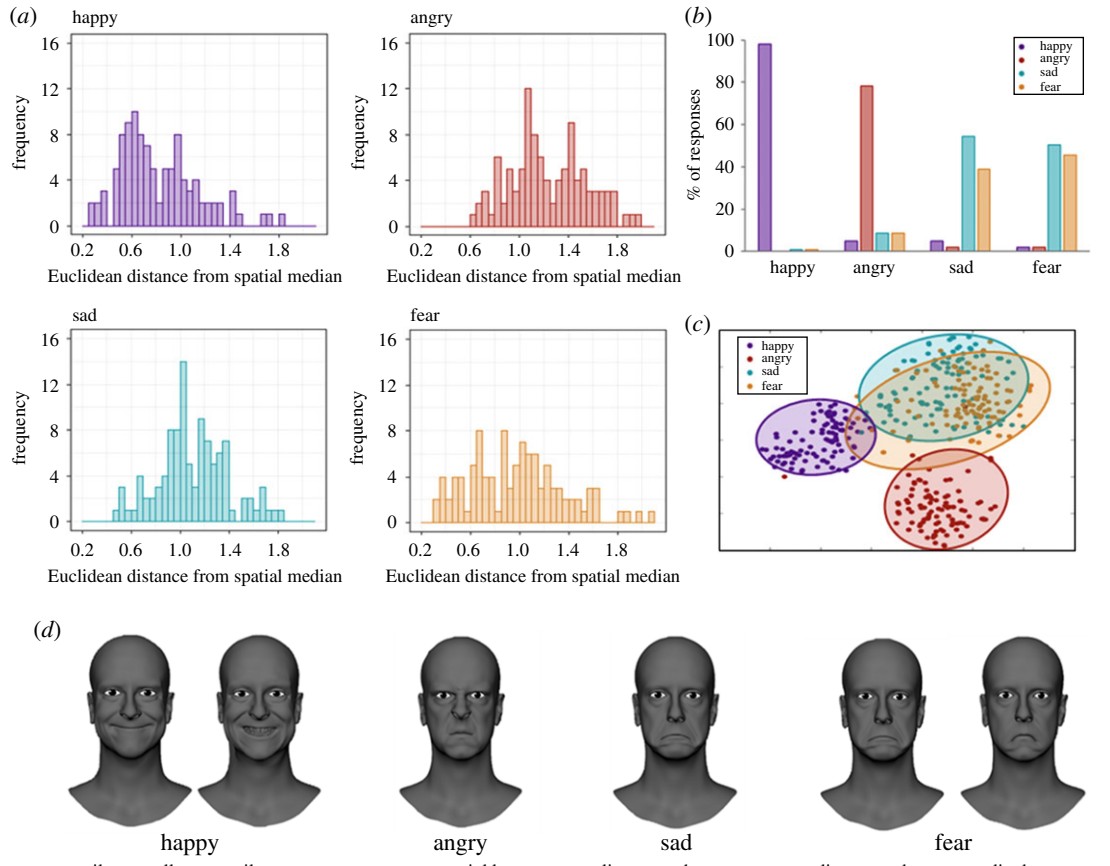

**Figure 2.** (*a*) Histogram plots showing the population distribution of Euclidian distances from the spatial median for each emotion category. A smaller distance represents less variation from the population norm. (*b*) For prototype faces generated within each emotion category, per cent of the total number of responses ($N = 105$) where the distance to the spatial median was closest to each respective emotion's spatial median. (*c*) Results of the t-stochastic neighbour embedding (t-SNE) data reduction and cluster visualization ($N = 105$). Two-dimensional spatial location of the clusters is arbitrary except relative to one another. (*d*) For illustrative purposes, neutral rig examples displaying the fullest expression of only the blendshape which was deemed most important in forming happy, angry, sad and fearful expressions. Blendshape weights for these features were set to 1, while all other blendshape weights in the rig were kept at 0 (neutral).

[$F_{1,358} = 48.05$, $p < 0.001$, adjusted $R^2 = 0.12$]. There was no significant difference in the degree of deviation from the norm for prototype angry faces compared with sad [$F_{1,358} = 2.75$, $p = 0.10$, adjusted $R^2 = 0.005$]. All significant findings survive Bonferroni correction for multiple comparisons at an adjusted threshold of $p = 0.05/6 = 0.0083$.

Euclidian distances obtained from multivariate dispersion analysis were used to observe, within each emotion category, the number of instances where the distance of an individual data point was closer to the spatial median of a different category than it was to the spatial median of its own category (figure 2*b*). This was interpreted as an index of 'confusability' among prototype faces generated for each emotion. The more instances where the distance of a prototype face was closer to the spatial median of a category other than its own, the higher degree of 'confusability' with other emotions among prototype faces from that category. Happy was the most consistent category, with 103 of 105 datapoints closest to the spatial median of happy (1 face was closer to sad, 1 closer to fear, 0 closer to angry). Angry was the next most consistent category, with 82 of 105 distances closest to the spatial median for angry (9 closer to sad, 9 closer to fear, 5 closer to happy). Sad and fear showed higher levels confusability with one another, with roughly half of the data points for one of these emotions being spatially closer to the median of the other (sad: 41 closer to fear, 5 closer to happy, 2 closer to angry; fear: 53 closer to sad, 2 closer to happy, 2 closer to angry).

Reducing the 42-dimensional face-space into two-dimensional space, t-SNE depicted participants' responses as a single metric clustered by the four emotion categories (figure 2*c*). Among these clusters, happy and angry were the most separable and least variable categories, forming distinct

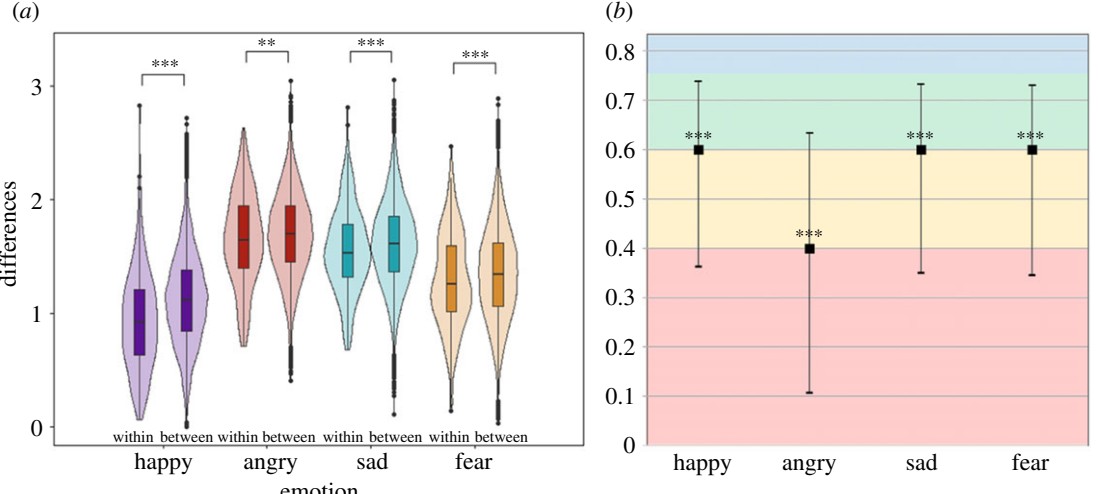

**Figure 3.** (*a*) Violin plots illustrating results of Mantel tests within each emotion category, comparing dissimilarity matrices of within- versus between-subject responses. Within each emotion category, left plots represent within-subject differences across the three time points, and right plots represent between-subject differences. Boxplots represent the median and interquartile range. *** indicates significant at $p < 0.0001$, ** indicates significant at $p < 0.001$. Plots indicate that between-subject variability was greater than within-subject variability across time for all four emotions. (*b*) Intraclass correlation coefficients for distances from the spatial median measured across the three time points. *** indicates significant at $p < 0.001$. Error bars represent standard error. Red = poor reliability, orange = fair reliability, green = good reliability, blue = excellent reliability [43].

clusters with little overlap among the other emotion categories. Sad and fear exhibited greater overlap compared with happy or angry, and these two categories appeared less tightly clustered (more diffuse) in two-dimensional space relative to happy and angry, suggesting there was greater variability among these prototype faces. While degrees of between-subject variability differed between emotions, the GA did allow the generation of faces that fell within reasonable norms of typical expressions for each emotion (see exemplars presented in electronic supplementary material, eFigure-2).

Figure 2*d* shows the blendshapes that were most important in forming happy, sad and fearful expressions, all of which involved mouth-related features, primarily those determining curvature of the mouth. Happy was most strongly linked to changes in *smile controller* (mean weight $0.29 \pm 0.26$) and *smile open* (mean weight $0.39 \pm 0.36$), although it is important to note that the mean weights for these blendshapes were less than 0.5, indicating that on average, a threshold of less than 50% of the full expression of these features was required to form the prototype happy expression. Sad was associated with *lip corner depressor* (mean weight $0.44 \pm 0.37$). Fear was also linked to *lip corner depressor* (mean weight $0.30 \pm 0.30$) as well as *lip down* (mean weight $0.47 \pm 0.32$). Only angry faces prioritized change in a feature other than the mouth (*nose wrinkler* (mean weight $0.83 \pm 0.24$)), with a comparatively higher weight for this feature required to generate a prototype fearful expression.

## 3.2. Intra-individual reliability

Mantel tests (figure 3*a*) showed that faces generated for all four emotions were positively correlated with the individual-identity matrix (happy: $\rho = 0.048$, $p < 001$; sad: $\rho = 0.018$, $p < 0.001$; angry: $\rho = 0.010$, $p < 0.01$; fear: $\rho = 0.016$, $p < 0.001$), indicating that variation between subjects was greater than within-subject variability across the three time points. Moreover, there was no significant difference of correlations in each participant between time point 1 and time point 2, time point 2 and time point 3 or time point 1 and time point 3 for any emotion, indicating that the amount of time between visits (30 min versus four to six weeks) did not significantly impact within-subject variability.

ICCs calculated across the three time points using distances from the spatial median indicated moderate to good response reliability for all four emotion categories: happy: ICC = 0.585, 95% CI = 0.363–0.739, $F_{59,118} = 2.410$, $p < 0.001$; sad: ICC = 0.576, 95% CI = 0.350–0.733, $F_{59,118} = 2.361$, $p < 0.001$; angry: ICC = 0.419, 95% CI = 0.107–0.634, $F_{59,118} = 1.720$, $p < 0.01$; fear: ICC = 0.574, 95% CI = 0.345–0.731, $F_{59,118} = 2.345$, $p < 0.001$ (figure 3*b*).

## 3.3. External validity

The most common external responses to prototype faces within each emotion category are displayed in figure 4a. The two most common responses for happy, angry and sad were exact matches for the emotion displayed. The 10 most common responses for happy faces were all synonymous with happy, suggesting high population-level agreement for happy prototype faces. The most common response to fearful faces was 'sad', and 'disgust' was among the 10 most common responses for angry, sad and fearful faces, indicating a higher degree of 'confusability' among these expressions, particularly fear.

Of 11 686 responses collected from external validators across all four emotion categories, 7752 (66.3%) were classed as 'correct', of which 5375 (69.3% of correct responses, 46% of total responses) were deemed an exact match; 3934 responses (33.7%) were classed as incorrect, of which 116 responses (2.9% of incorrect responses, 1% of total responses) were classed as anomalous. Accuracy rates per emotion are displayed in figure 4b. Happy faces had the highest accuracy rate, at 91.8% (68.8% of which were an exact match); 81.3% of angry faces were correctly identified (75.2% of which were an exact match) and 60.8% of sad faces (80.4% of which were an exact match). Fearful faces had the lowest accuracy rate, at 31.0% (35.6% of which were an exact match). Of total incorrect responses for each emotion category, 12.3% were deemed anomalous for happy faces, 4.5% for angry faces, 2.5% for sad faces and 1.7% for fearful faces.

## 4. Discussion

Combining methods from computer animation and psychophysics, we developed a GA task which identified individual differences in the way people perceive and internally represent facial emotional expressions, addressing a gap in the field of emotion research [9]. The results from this study are manifold: first, we showed that the GA paradigm can efficiently generate reliable individualized prototype facial expressions that are consistent across time. Second, we found that within the population, people are more consistent in their representations of happy faces compared with angry, fearful and sad faces. Third, we observed that an independent sample of individuals was able to identify emotions displayed by the prototype expressions generated from the GA, but that this ability was more accurate and less variable for positive (i.e. happy) relative to negative emotions (particularly fear). Collectively, these findings suggest that it is possible to generate personalized, quantifiable stimuli to highlight individual differences in emotion representation. This study also provides preliminary evidence that many of the commonly employed paradigms using standardized sad and fearful stimuli may not be adequately capturing individual differences in how people perceive and represent these emotions.

The reliability and generalizability of traditional paradigms used to measure emotion processing has recently been called into question [9,44,45]. Advances in data-driven approaches [2,24,32–34] have worked towards developing individualized measures of emotion processing, but these methods are often limited by their requirement of many trials, making them unsuitable for use in large or specialist study groups such as children or clinical populations. By contrast, our study shows that GAs are a reliable and efficient alternative that can generate adaptable, individualized representations of facial emotion in a matter of minutes. We not only showed that there was less variability within individuals across time compared with between participants, but we also observed a high degree of reliability of task measures, confirming the ability of this task to distinguish within- versus between-subject performance. This characteristic further underscores the utility of this method for the investigation of individual differences, presenting scope to distinguish between individuals within a population. Moreover, the external validation study showed that participants were able to correctly identify two-thirds of overall responses, in line with previous work showing average accuracy rates for emotion recognition between 64% and 67% [46].

Clustering results showed that prototype angry and happy faces were distinct from the other emotions, and these two emotions had the highest identification accuracy rates in external validation. This finding for angry faces is in line with previous work showing that the detection of angry faces is faster and more efficient compared with other emotions [47]. Similarly, results for happy support the hypothesized '*happy face superiority effect*', suggesting that happiness is recognized faster and more accurately compared with other emotions as an evolutionary mechanism to promote group cohesion and social bonds [46,48,49]. It has also been suggested that humans are able to more readily identify happy faces because we are more frequently exposed to happy expressions in our everyday

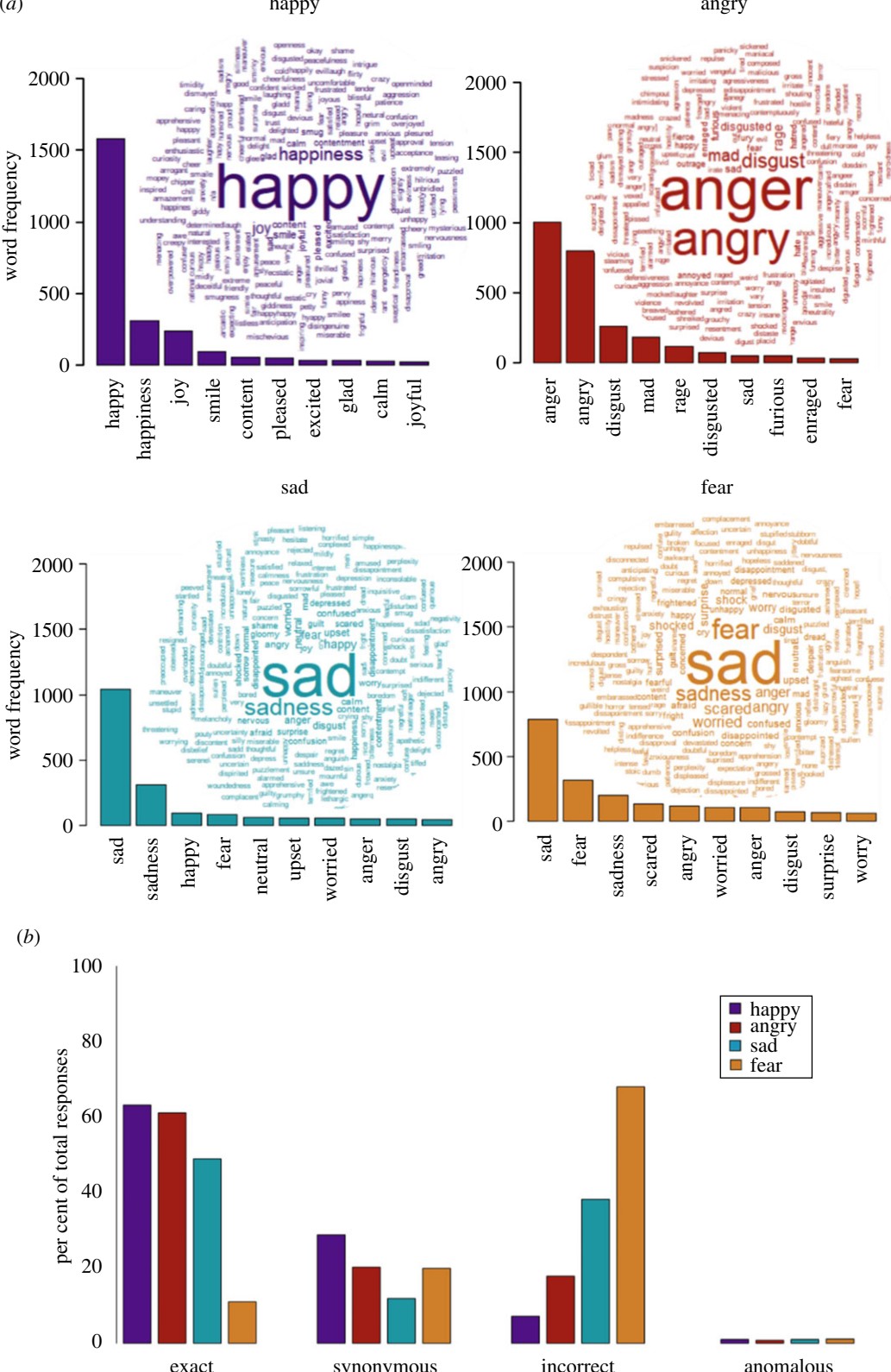

**Figure 4.** (*a*) Word cloud and histogram representations of the most common responses among online external validator participants within each emotion category when asked to respond to randomly selected prototype faces created from the GA task using one word in a free-text response box. Histograms depict the ten most commonly provided responses, ranked from most to least common. (*b*) Bar chart depicting proportion of correct (exact and synonymous) and incorrect (incorrect and anomalous) responses made by online external validator participants.

environments compared with, e.g. fearful expressions, highlighting a hypothetical 'training effect' of stimulus exposure on recognition accuracy [50]. Lastly, the fact that angry and happy prototype faces required expression of blendshapes in facial movements distinct from those needed for fear and sad expressions may clarify the distinct mechanisms driving differences in representations across emotions.

There was a high degree of overlap for the representation and perception of fearful and sad faces. Fearful and sad expressions had more features in common compared with happy or angry expressions, and external validation showed that these two emotions were least consistently correctly identified, with fearful expressions most often perceived as sad. This is in line with findings of poorer classification accuracy and more frequent confusion among fearful and sad expressions relative to happiness and anger [51,52]. Overlap and poorer identification for sad and fear are likely to reflect the fact that these expressions are inherently more ambiguous and complex [53], more context-dependent than happiness and anger [54], and are more likely to co-occur in daily life. Previous work has also shown that both sad and fearful expressions are associated with the action unit of stretched mouth [55], which parallels our findings of lip corner depressor being a key blendshape for sad and fearful faces. Additionally, people respond with lower accuracy when identifying fearful expressions compared with other basic emotions [56], an effect that is thought to be due to the fact that fearful facial expressions are encountered less frequently in our everyday lives [57]. Moreover, it has been argued that there is a social need to express happiness more explicitly to foster social cohesion, while other emotions (e.g. fear, sadness) may be camouflaged to avoid exposing vulnerability or distress [58]. This innate social pressure may in part contribute to the 'confusability' observed between fear and sadness. Importantly, our study shows that this 'confusability' effect is present when perceiving personalized, rather than standardized, representations of emotion. It is also worth noting that in our test–retest reliability analysis, while all emotions had relatively high reliability, angry faces had lower reliability compared with other emotions, yet angry faces were more often accurately recognized compared with fear and sad. With regard to the validity of fearful and sad faces, these results speak to the fact that participants are consistently producing the same faces for these emotions (their representation is stable), but it is instead others' interpretation of these expressions which is variable.

This study has many strengths. To our knowledge, this is the first application of GAs to facial expression generation, highlighting the promise of these methods for capturing individual differences in how humans represent and perceive emotional expressions. For example, many studies have used emotional stimuli such as fearful faces to examine the associations between the neurocognitive processing of emotion and the presence of behavioural or psychiatric traits [59–61]. These studies have assumed that the emotion depicted by the standardized stimuli is comparably represented by a diverse group of individuals. Yet we know that emotion perception varies at the individual level, and there have been recent calls to refine the measurement of this variation [9,62,63]. Our results highlight the importance of the sensitivity of stimuli in emotion processing paradigms, as there may be a large degree of population-level variability (especially with regard to fear) in the way that individuals perceive and respond to emotional faces. A key benefit of our approach is the ability to directly quantify this variability and investigate its impact on wide-ranging aspects of emotion processing, from how we perceive the expressions of others to how we express our own emotions. An index of the variance of expressions generated by different participants within emotion categories may prove especially useful in psychopathology research where greater or less variation within a category could identify departures from the population norm and association with problematic behavioural traits and symptoms. Ultimately, the evidence from our study indicates that improvements in how we measure variation could be of great value in understanding mechanistic underpinnings of both normal and abnormal emotion processing, elucidating why some people fail to respond appropriately to emotional expressions.

## 4.1. Limitations and future work

This initial study has paved the way for refinement of the GA approach in future studies but is not without limitations. The faces used in the current interface are computer-generated rather than photo-realistic. While these faces approximate portrayals of real human faces, their representation still differs from 'real-world' stimuli [64]. There are very few open-source artistic rigs available containing correctives to avoid unrealistic shapes in a user-guided GA. Future refinements to our method could add additional constraints to avoid unrealistic faces in cases where blendshapes do not have correctives. Alternatively, given that our rig was governed by the FACS [7], our method presents scope for future replication, as studies could apply the GA method to other FACS-based rigs and

should also aim to validate the generation of expressions of other proposed categorical emotions such as e.g. surprise and disgust [65]. Moreover, future iterations of this task will leverage higher fidelity photogrammetric and photographic facial data together with recent advances in neural rendering [66]. We also acknowledge the importance of context in the representation, perception and interpretation of emotional expressions [54,67], despite the fact that much emotion research operates on the assumption that basic emotion categories are universally recognized and perceived as intended by the experimenter. Our current task partially addresses this limitation by allowing subjects to generate representations of their own ideals of facial emotions, but future studies should examine these representations across various stimulus-based, perceiver-based and cultural contexts which probably impact the encoding and perception of facial expressions. It has also been suggested that morphologically similar expressions such as sadness and fear may differ primarily in their dynamics (i.e. facial movements) [68]. Comparing GA-generated dynamic stimuli could prove a fruitful future step in isolating the source of individual differences. The high within-subject reliability suggests that the GA used an appropriate number of iterations for adequate convergence, and mitigates the likelihood that the GA fell into a local minimum. Moreover, across time, the fact that participants generated faces that were more similar to each other than they were to faces generated by other participants suggested that the faces produced by the GA were valid individualized representations of emotion; the GA was able to capture individual variability in the way we represent emotional expressions. However, future studies should empirically investigate whether there was sufficient convergence of the GA after six generations. Future work should also examine the test–retest reliability of this task using a different initial population of test faces at each time point. Lastly, in addition to comparing blendshape weights, comparing geometry of the meshes generated for each participant/emotion would further add to the interpretation of our findings.

This task has promising implications for developmental and psychopathology research that was beyond the scope of this initial study. For example, there is evidence of differences in emotion perception based on age [69,70], sex [71,72] and symptoms associated with psychopathology [73–75]. Understanding the relationship between these traits and individual differences in emotion representation can shed light on whether there are biases related to certain emotions in specific populations, or whether the threshold needed to produce or respond to an emotionally salient expression is higher or lower in some individuals.

Ultimately, the task and findings presented in this study represent progress in the measure of individual differences in emotion processing, with future applications for both developmental and clinical research. Findings corroborate existing evidence for separable effects of the four emotion categories investigated, while highlighting emotion-specific patterns of variability that differ across categories.

Ethics. All participants gave written informed consent, and the study was approved by the UCL Research Ethics Committee (Z6364106/2013/07/37). Data were processed in accordance with EU General Data Protection Regulations.

Data accessibility. This paper is available online as a preprint, doi:10.31234/osf.io/9fta2. Data and code are available on figshare (doi:10.6084/m9.figshare.11743308; doi:10.6084/m9.figshare.11743281; doi:10.6084/m9.figshare.11743326; doi:10.6084/m9.figshare.11743716).

Authors' contributions. C.O.C., D.P.C., E.V. and I.M. conceived of and designed the study. K.R. developed the study paradigm. C.O.C. and F.G.L.H. collected the data. C.O.C., F.G.L.H., K.R. and R.L. analysed and interpreted the data. C.O.C. wrote the paper, and all authors edited the work and approved the final version to be submitted for publication.

Competing interests. At the time of writing, Prof. Essi Viding and Prof. Isabelle Mareschal are Board Members of Royal Society Open Science, but had no involvement in the review or assessment of the paper. All other authors declare no competing interests.

Funding. C.O.C. received support as a Wellcome Trust Postdoctoral Fellow (grant no. 206459/Z/17/Z). K.R.'s doctoral studentship was funded by the EPSRC (grant no. EP/L016540/1). D.P.C., E.V. and I.M. received support from the MRC (grant no. MR/S011307/1). D.P.C. directs the Centre for the Analysis of Motion, Entertainment Research and Applications (EPSRC EP/M023281/1). E.V. was supported by a British Academy Mid-Career Fellowship.

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
