## [Peer Review File · Royal Society Open Science]

Review History

RSOS-202251.R0 (Original submission)

Review form: Reviewer 1

Is the manuscript scientifically sound in its present form?

No

Are the interpretations and conclusions justified by the results?

No

Is the language acceptable?

Yes

Do you have any ethical concerns with this paper?

No

Have you any concerns about statistical analyses in this paper?

Yes

Recommendation?

Major revision is needed (please make suggestions in comments)

Comments to the Author(s)

The authors provide data that are meant to capture idiosyncrasies in expressing basic emotions using genetic algorithms.

I have a number of questions and concerns.

- Is this paper really about individual differences or about idiosyncrasies? The authors seem to think it is about individual differences but to me it seems to be more about idiosyncrasies and peculiarities of persons.
- The genetic algorithm is just a meta heuristic that can be applied to some computational problems. I do not think is specifically suited to the current research questions - at least not more so than the application of competing meta-heuristics. If this paper is meant as an endorsement of a specific meta-heuristic, the authors need to make a better case. Else they should adjust title, abstract, and paper to make it clear what this paper is really about.
- The choice of 4 emotions is poorly justified. Why these but not say disgust, contempt, surprise etc.?
- The paragraph on page 6 lines 28 to 42 is hard to understand. A GA task was created? I though the authors would apply a genetic algorithm to expressions subjects provide? What does the no-constraints statement mean? Subjects are verbally prompted to express one out of four emotions aren't they? Isn't that pretty constrained (i.e. do what experimenter asks you to do)? How and which choices facilitate stimulus creation? I guess this whole paragraph needs rewriting, replacement, reconsideration.
- The description of the procedure/method is not satisfactory at all. Critical aspects are provided elsewhere. The facial features are action units? The choices of features and how they change over iterations, the noise and mutations added all need to be described in an accessible way within this publication.
- Am I right in understanding the design stating that participants always only saw pictures of themselves (along with some machine generated modifications [according to action units selected by participants as essential for showing one emotion or another])?
- The power/sample size analysis seems to be for a purpose at best tangential to the present research questions. This section should also appear elsewhere and it should be tailored to the present research questions, preferably as a priori power analysis.
- In section 4.1. the authors begin with detailing their procedure for computing the discrepancy between the preferred participant face for a specific emotion and some 42 dimensional face-space. Much more detail on this computation is required.
- From figure two it seems as if lip corner depressor is clearly the most distinctive feature in both fear and sadness, relative to other action units. May be this is decisive for the issues in distinguishing between sad and fear and I doubt this is what the authors (should have) wanted.
- It wasn't easy to decode the design and it was harder still to try to make sense out of the decisions made by the authors. The text might have a much better flow, if the research question were made absolutely explicit. This could be for example the idiosyncratic nature of emotion expression across individuals. It could also be the use of GA to help participants approximate their personal emotion expression space. In the current version of the text both seem to be the case (and obviously there are a number of additional research questions the authors could pursue).
- The purpose of the external validity sample seems unclear to me. Something that is quite obscure at this stage is arguably validated with something that is even more obscure. The data also seem to be insufficiently cleared from nonsense observations.
- I am afraid this paper will need substantial revision and may be additional data. I like the gist of using GA for altering, modifying facial stimuli. Machine-based face generation might also be

used to help subjects approximate their personal expression space. If this were part of the research questions, adequate display/visualization of such idiosyncrasies is essential.

Review form: Reviewer 2

Is the manuscript scientifically sound in its present form?

No

Are the interpretations and conclusions justified by the results?

No

Is the language acceptable?

Yes

Do you have any ethical concerns with this paper?

No

Have you any concerns about statistical analyses in this paper?

No

Recommendation?

Major revision is needed (please make suggestions in comments)

Comments to the Author(s)

The paper presents an interesting idea, but it raises some concerns in its present form that the authors should address, particularly the following.

The use of Genetic Algorithms (GAs) can provide a promising approach to study individual differences in (facial) emotion generation and perception. However, the use of interactive GAs to generate "personalized" emotional facial expressions is not new, and studies can be found for example in the fields of artificial life, artificial intelligence and animation since the mid-90s, which range in the types of faces and expressions used from the highly streamline drawings to morphing technology to 3D realistic animations. Moreover, recent work has used generative adversarial networks to generate contextually-relevant facial expressions in dyadic human interactions. The authors should discuss such related work, both in terms of the interactive generation of individualized expressions and of the GAs / algorithms used.

More details of the GA used and justification of the choices made should be discussed in the body of the paper, since the choice of the GA and its operators can have a big impact on the results obtained.

However, my main concerns related to the fact that it is not clear which is the point the authors are trying to make and how the study addresses it. First, the authors criticize the use of "categorical" models and studies, as they overlook individual differences; however, the present study uses a categorical approach, which seems contradictory. Would the authors be criticizing the use of categories or the universality claim? In any case, categorical/universal approaches do take into account individual differences in emotion expression and perception, and more recent studies (including by Ekman) can be found comparing "normative" aspects and individual differences (see for example recent publications by Ekman). Second, it is not clear what the contribution of this study to the study of individual differences is; large parts of the discussion focus on differences across expressions (categories) rather than trying to shed some light on

individual differences, their context, and their significance. Finally, much like the "decontextualized" studies that the authors criticize, this study overlooks the role of the context in the expression, perception and interpretation of emotional expression, which is one of the key elements stressed by authors like Barrett-Feldman, which the authors quote to support the idea of the importance of individual differences.

Decision letter (RSOS-202251.R0)

Dear Dr Carlisi

We have received comments from reviewers regarding your paper RSOS-202251 "Using Genetic Algorithms to uncover individual differences in how humans represent facial emotion" and would like you to revise the paper in accordance with these comments, please. Please note this decision does not guarantee eventual acceptance.

We invite you to respond to the comments supplied below and revise your manuscript. Below the referees' comments we provide additional requirements for the preparation of your revision. Final acceptance of your manuscript is dependent on these requirements being met. We provide guidance below to help you prepare your revision.

Please submit your revised manuscript and required files (see below) no later than 21 days from today's (ie 03-Mar-2021) date. Note: the ScholarOne system will 'lock' if submission of the revision is attempted 21 or more days after the deadline. If you do not think you will be able to meet this deadline please contact the editorial office immediately.

Please note article processing charges apply to papers accepted for publication in Royal Society Open Science (<https://royalsocietypublishing.org/rsos/charges>). Charges will also apply to papers transferred to the journal from other Royal Society Publishing journals, as well as papers submitted as part of our collaboration with the Royal Society of Chemistry (<https://royalsocietypublishing.org/rsos/chemistry>). Fee waivers may be available but must be requested when you submit your revision (<https://royalsocietypublishing.org/rsos/waivers>).

Reviewer comments to Author:

Reviewer: 1

Comments to the Author(s)

The authors provide data that are meant to capture idiosyncrasies in expressing basic emotions using genetic algorithms.

I have a number of questions and concerns.

- Is this paper really about individual differences or about idiosyncrasies? The authors seem to think it is about individual differences but to me it seems to be more about idiosyncrasies and peculiarities of persons.
- The genetic algorithm is just a meta heuristic that can be applied to some computational problems. I do not think is specifically suited to the current research questions - at least not more so than the application of competing meta-heuristics. If this paper is meant as an endorsement of a specific meta-heuristic, the authors need to make a better case. Else they should adjust title, abstract, and paper to make it clear what this paper is really about.
- The choice of 4 emotions is poorly justified. Why these but not say disgust, contempt, surprise etc.?
- The paragraph on page 6 lines 28 to 42 is hard to understand. A GA task was created? I though the authors would apply a genetic algorithm to expressions subjects provide? What does the no-constraints statement mean? Subjects are verbally prompted to express one out of four emotions aren't they? Isn't that pretty constrained (i.e. do what experimenter asks you to do)? How and which choices facilitate stimulus creation? I guess this whole paragraph needs rewriting, replacement, reconsideration.
- The description of the procedure/method is not satisfactory at all. Critical aspects are provided elsewhere. The facial features are action units? The choices of features and how they change over iterations, the noise and mutations added all need to be described in an accessible way within this publication.
- Am I right in understanding the design stating that participants always only saw pictures of themselves (along with some machine generated modifications [according to action units selected by participants as essential for showing one emotion or another])?
- The power/sample size analysis seems to be for a purpose at best tangential to the present research questions. This section should also appear elsewhere and it should be tailored to the present research questions, preferably as a priori power analysis.
- In section 4.1. the authors begin with detailing their procedure for computing the discrepancy between the preferred participant face for a specific emotion and some 42 dimensional face-space. Much more detail on this computation is required.
- From figure two it seems as if lip corner depressor is clearly the most distinctive feature in both fear and sadness, relative to other action units. May be this is decisive for the issues in distinguishing between sad and fear and I doubt this is what the authors (should have) wanted.
- It wasn't easy to decode the design and it was harder still to try to make sense out of the decisions made by the authors. The text might have a much better flow, if the research question were made absolutely explicit. This could be for example the idiosyncratic nature of emotion expression across individuals. It could also be the use of GA to help participants approximate their personal emotion expression space. In the current version of the text both seem to be the case (and obviously there are a number of additional research questions the authors could pursue).
- The purpose of the external validity sample seems unclear to me. Something that is quite obscure at this stage is arguably validated with something that is even more obscure. The data also seem to be insufficiently cleared from nonsense observations.
- I am afraid this paper will need substantial revision and may be additional data. I like the gist of using GA for altering, modifying facial stimuli. Machine-based face generation might also be used to help subjects approximate their personal expression space. If this were part of the research questions, adequate display/visualization of such idiosyncrasies is essential.

Reviewer: 2

Comments to the Author(s)

The paper presents an interesting idea, but it raises some concerns in its present form that the authors should address, particularly the following.

The use of Genetic Algorithms (GAs) can provide a promising approach to study individual differences in (facial) emotion generation and perception. However, the use of interactive GAs to generate "personalized" emotional facial expressions is not new, and studies can be found for example in the fields of artificial life, artificial intelligence and animation since the mid-90s, which range in the types of faces and expressions used from the highly streamline drawings to morphing technology to 3D realistic animations. Moreover, recent work has used generative adversarial networks to generate contextually-relevant facial expressions in dyadic human interactions. The authors should discuss such related work, both in terms of the interactive generation of individualized expressions and of the GAs / algorithms used.

More details of the GA used and justification of the choices made should be discussed in the body of the paper, since the choice of the GA and its operators can have a big impact on the results obtained.

However, my main concerns related to the fact that it is not clear which is the point the authors are trying to make and how the study addresses it. First, the authors criticize the use of "categorical" models and studies, as they overlook individual differences; however, the present study uses a categorical approach, which seems contradictory. Would the authors be criticizing the use of categories or the universality claim? In any case, categorical/universal approaches do take into account individual differences in emotion expression and perception, and more recent studies (including by Ekman) can be found comparing "normative" aspects and individual differences (see for example recent publications by Ekman). Second, it is not clear what the contribution of this study to the study of individual differences is; large parts of the discussion focus on differences across expressions (categories) rather than trying to shed some light on individual differences, their context, and their significance. Finally, much like the "decontextualized" studies that the authors criticize, this study overlooks the role of the context in the expression, perception and interpretation of emotional expression, which is one of the key elements stressed by authors like Barrett-Feldman, which the authors quote to support the idea of the importance of individual differences.

===PREPARING YOUR MANUSCRIPT===

===PREPARING YOUR REVISION IN SCHOLARONE===

-- Ensure that your data access statement meets the requirements at <https://royalsociety.org/journals/authors/author-guidelines/#data>. You should ensure that you cite the dataset in your reference list. If you have deposited data etc in the Dryad repository, please include both the 'For publication' link and 'For review' link at this stage.

-- If you have uploaded ESM files, please ensure you follow the guidance at <https://royalsociety.org/journals/authors/author-guidelines/#supplementary-material> to include a suitable title and informative caption. An example of appropriate titling and captioning may be found at https://figshare.com/articles/Table_S2_from_Is_there_a_trade-off_between_peak_performance_and_performance_breadth_across_temperatures_for_aerobic_sc_ope_in_teleost_fishes_/3843624.

Author's Response to Decision Letter for (RSOS-202251.R0)

See Appendix A.

RSOS-202251.R1 (Revision)

Review form: Reviewer 1

Is the manuscript scientifically sound in its present form?

No

Are the interpretations and conclusions justified by the results?

Yes

Is the language acceptable?

Yes

Do you have any ethical concerns with this paper?

No

Have you any concerns about statistical analyses in this paper?

No

Recommendation?

Major revision is needed (please make suggestions in comments)

Comments to the Author(s)

I think a strength of the present paper is that a novel tool to study subjective prototypes of expressions of emotion categories is developed. However, the validation of this tool is not

presented in a coherent form. It seems my earlier review was not perceived as very helpful. But if I reconsider my concerns they have mostly not changed in this revision.

My first concern was:

- Is this paper really about individual differences or about idiosyncrasies? The authors seem to think it is about individual differences but to me it seems to be more about idiosyncrasies and peculiarities of persons.

Apparently, the tool the authors use might have the potential to show that some face-space of a person is unique/distinct or deformed. These would be idiosyncrasies of individual subjects. There are many exciting research questions that could be studied that way (say dynamic factor analysis of individual face spaces within subjects) Of course individual peculiarities can be studied across participants too. But neither research approach is pursued convincingly.

My second concern was about GA being a meta-heuristic - one amongst many. If the key contribution of this paper is computational I might not be a good reviewer - but I think it is evident that competing approaches should be discussed and considered. Instead the authors argue I haven't demonstrated that GA is suboptimal. But is this my task as a reviewer? I don't think so.

I see a number of options what this paper is all about, but none of them is pursued convincingly.

Let me illustrate this point. The abstract states: However, emotion research to date has generally relied on the assumption that people represent categorical emotions in the same way, using standardised stimulus sets and overlooking important individual differences. To resolve this problem, we developed and tested a task using genetic algorithms to derive assumption-free, participant-generated emotional expressions.

Inferring from this statement one would expect that the present set of stimuli are juxtaposed to established sets of stimuli and the superiority of the present set is shown one way or another. It might be seen as partisan that the universality of perceiving emotion expressions is rejected ex cathedra. Obviously, deviations from such a universal model could be quantitative or qualitative and it is not exactly clear which sort of deviation the present data support.

In a sense the paper might be overburdened by trying to solve to many issues at a time. It could be sufficient to compare participant-generated expressions with a number of machine classified expressions within and between subjects and to further study any emerging differences then.

Review form: Reviewer 2

Is the manuscript scientifically sound in its present form?

Yes

Are the interpretations and conclusions justified by the results?

Yes

Is the language acceptable?

Yes

Do you have any ethical concerns with this paper?

No

Have you any concerns about statistical analyses in this paper?

No

Recommendation?

Accept as is

Comments to the Author(s)

I am happy with the modifications made following the first review

Decision letter (RSOS-202251.R1)

Dear Dr Carlisi,

I am pleased to inform you that your manuscript entitled "Using Genetic Algorithms to uncover individual differences in how humans represent facial emotion" is now accepted for publication in Royal Society Open Science. The comments of the Editor are below.

Associate Editor Comments to Author (Prof César Lima):

I agree with Reviewer 1 that the main strength of this work is that it implements a new approach to study variability in the representation of facial emotional expressions. Regarding the raised concerns (individual differences vs. idiosyncrasies; consideration of other computational approaches), they have already been raised in the previous round, and the authors have made significant changes to address them. However, the report primarily focusses on restating the concerns, rather than on evaluating the authors' response or providing specific suggestions for improvement.

An additional critical point is raised, suggesting that the authors aim to reject the universal model of emotion without providing supporting data or explaining whether the changes are qualitative or quantitative. It is not clear to me what qualitative vs. quantitative mean in this context, and my view on the authors' main point is distinct. They do allude to the current debate about whether basic emotions are universally recognized/produced, but their argument is about explaining how their methodological approach highlights individual differences, more than about defending a specific theoretical perspective or solving the debate.

Reviewer comments to Author:

Reviewer: 1

Comments to the Author(s)

I think a strength of the present paper is that a novel tool to study subjective prototypes of expressions of emotion categories is developed. However, the validation of this tool is not presented in a coherent form. It seems my earlier review was not perceived as very helpful. But if I reconsider my concerns they have mostly not changed in this revision.

My first concern was:

- Is this paper really about individual differences or about idiosyncrasies? The authors seem to think it is about individual differences but to me it seems to be more about idiosyncrasies and peculiarities of persons.

Apparently, the tool the authors use might have the potential to show that some face-space of a person is unique/distinct or deformed. These would be idiosyncrasies of individual subjects. There are many exciting research questions that could be studied that way (say dynamic factor analysis of individual face spaces within subjects) Of course individual peculiarities can be studied across participants too. But neither research approach is pursued convincingly.

My second concern was about GA being a meta-heuristic - one amongst many. If the key contribution of this paper is computational I might not be a good reviewer - but I think it is evident that competing approaches should be discussed and considered. Instead the authors argue I haven't demonstrated that GA is suboptimal. But is this my task as a reviewer? I don't think so.

I see a number of options what this paper is all about, but none of them is pursued convincingly.

Let me illustrate this point. The abstract states: However, emotion

research to date has generally relied on the assumption that people represent categorical emotions in the same way, using standardised stimulus sets and overlooking important individual differences. To resolve this problem, we developed and tested a task using genetic algorithms to derive assumption-free, participant-generated emotional expressions.

Inferring from this statement one would expect that the present set of stimuli are juxtaposed to established sets of stimuli and the superiority of the present set is shown one way or another. It might be seen as partisan that the universality of perceiving emotion expressions is rejected ex cathedra. Obviously, deviations from such a universal model could be quantitative or qualitative and it is not exactly clear which sort of deviation the present data support.

In a sense the paper might be overburdened by trying to solve to many issues at a time. It could be sufficient to compare participant-generated expressions with a number of machine classified expressions within and between subjects and to further study any emerging differences then.

Reviewer: 2

Comments to the Author(s)

I am happy with the modifications made following the first review

Appendix A

30th March, 2021

Dear Professor Brazil,

Thank you for the opportunity to revise and resubmit our manuscript, "Using Genetic Algorithms to uncover individual differences in how humans represent facial emotion." We are grateful for the opportunity to address the reviewers' comments. Below, we provide a detailed response to each reviewer in turn (reviewer comments in black, our responses in blue). We have also included two versions of the manuscript in the resubmission, one 'clean' version, and one version which contains highlighted text that has been added or changed in response to reviewers' comments.

Thank you again for this opportunity, and we very much hope that our paper is now suitable for publication in *Royal Society Open Science*.

Yours sincerely,

Christina Carlisi, Darren Cosker, Essi Viding, Isabelle Mareschal

--

Reviewer: 1

Comments to the Author(s)

The authors provide data that are meant to capture idiosyncrasies in expressing basic emotions using genetic algorithms.

I have a number of questions and concerns.

- Is this paper really about individual differences or about idiosyncrasies? The authors seem to think it is about individual differences but to me it seems to be more about idiosyncrasies and peculiarities of persons.

As we interpret this comment, the reviewer is referring to 'idiosyncrasies' as a feature of task performance that fluctuates within an individual and thus not particularly informative about the individual, whereas 'individual differences' relate to enduring or stable characteristics within an individual that distinguish them from other individuals.

Based on this working assumption, we maintain that our study is a demonstration of individual differences in one's representation of emotional expressions. In particular, we would like to highlight the reliability analyses which were originally presented in the supplement but have now been moved to the main text (page 17). These analyses show that across three timepoints, individual responses show acceptable reliability, and that across time, a given participant's responses are more internally consistent (i.e. consistent with themselves) compared to other participants' responses. This indicates that any population-level variability observed in emotion representations is likely attributable to individual differences. Moreover, to our knowledge, the reliability and stability of responses regarding

subjective, individual emotion representation across time has not previously been investigated.

To further clarify the value of this within-subject reliability in delineating individual differences, we have added the following text to the discussion:

Page 20: *“We not only showed that there was less variability within individuals across time compared to between participants, but we also observed a high degree of reliability of task measures, confirming the ability of this task to distinguish within versus between-subject performance. This characteristic further underscores the utility of this method for the investigation of individual differences, presenting scope to distinguish between individuals within a population.”*

- The genetic algorithm is just a meta heuristic that can be applied to some computational problems. I do not think is specifically suited to the current research questions - at least not more so than the application of competing meta-heuristics. If this paper is meant as an endorsement of a specific meta-heuristic, the authors need to make a better case. Else they should adjust title, abstract, and paper to make it clear what this paper is really about.

We agree that there may be other computational approaches that could also be used to address our research questions, however the reviewer does not specify which competing meta-heuristics may be better-suited. The point we make in this paper is that we have developed a method that allows people to create their own, realistic and meaningful, emotional faces that are robust to fluctuations in internal representations (e.g., see point above regarding individual differences and reliability in our response to the previous comment). Our aim was to develop a method that was robust, realistic and efficient. The fact that there may be (many) other meta-heuristics does not mean that the one we chose is inappropriate. Nonetheless, we have added the following text to the introduction to highlight other approaches and to further clarify why we consider the genetic algorithm approach to hold particular promise over many other approaches:

Page 4: *“Participant-generated facial expressions have been of interest to the computer animation community for some time (Ersotelos and Dong 2008). Since the introduction of performance-driven animation (Williams 1990), there have been numerous implementations of data-driven methods, including principle component analysis (PCA)(Banz and Vetter 1999; Chai and Hodgins 2005), Gaussian Process Latent Variable Models (GPLVMs)(Grochow et al. 2004; Wang, Fleet, and Hertzmann 2007), and autoencoders (Fragkiadaki et al. 2015; Holden, Saito, and Komura 2016) to create realistic, performance (i.e. participant) driven graphics (Shin and Lee 2009; Yu, Garrod, and Schyns 2012). However, these techniques often rely on the secondary judgement of a skilled artist and are limited in their ability to allow users to explore and interpret resulting graphics. More recently, machine learning techniques such as Generative Adversarial Networks (GANs)(Goodfellow et al. 2014; Khan et al. 2021) and verbal crowd-shaping (Streuber et al. 2016) have been applied to create computer-generated facial expressions and body images based on sampled parameters and descriptive labels from a population. Whilst promising, these methods rely on unsupervised learning and classification, requiring vast training datasets and fully trained network for*

optimal performance. Therefore, such approaches may not be best-suited to index the nuanced and subjective nature of faces and emotion representation...

We argue that the key benefit of the GA approach we have employed is its efficiency in deriving subjective, individualised (i.e. not standardised), and quantifiable representations of emotional expressions – which was a key aim of our study. Prior work has successfully used data-driven psychophysical methods such as reverse correlation (see e.g. Jack and Schyns, 2017, *Annu Rev Psychol*; 68:269-297) to visualise mental representations of facial expressions, but these approaches require a significant amount of time and large numbers of trials, somewhat limiting their application for cognitive neuroscience research in certain populations (e.g., children, clinically-vulnerable). Moreover, these approaches critically do not allow the *creation* of such representations, they simply provide a more detailed account of what parts of a face are used to recognise particular emotions. Our approach is fundamentally different and, we believe, is the best approach to balance efficiency and practicality with obtaining a suitably fine-grained index of participants' own mental representations, as we state in the following amended text:

Page 5: *“While some studies have used data-driven approaches with non-static stimuli (Chen et al. 2018; Jack and Schyns 2015; Rychlowska et al. 2017; Yu et al. 2015, 2012), these methods are time consuming, usually requiring a large number of trials, and therefore ill-suited for large-scale testing (or testing child or clinical populations). Moreover, a true depiction of an individual’s internal representation of facial expressions has traditionally relied on highly-skilled artists, as is frequently the case in computer animation. The GA presents a new heuristic which allows the generation of individuals’ internal representations of expressions without the lengthy process of artist renderings.”*

- The choice of 4 emotions is poorly justified. Why these but not say disgust, contempt, surprise etc.?

We agree that it is important to investigate other emotions with the GA approach we present. However, prior to this study, GAs had not been applied in the context of population-level facial expression generation, and we thought that it was parsimonious to use four widely researched ‘basic emotion’ categories (Lindquist et al., 2013) as a starting point. The purpose of this paper was to help us demonstrate that the GA approach offers a valid and informative methodology for generating and comparing expressions across individuals and across categories, which is why we chose the four basic emotion categories most commonly employed in psychological research on facial emotion (Jack et al., 2014) in this initial study.

It will indeed be important in future research to investigate other emotions, and we now highlight this in the discussion:

Page 24: *“Alternatively, given that our rig was governed by the FACS (Ekman, 1978), our method presents scope for future replication, as studies could apply the GA method to other FACS-based rigs and should also aim to validate the generation of expressions of other proposed categorical emotions such as e.g., surprise and disgust (Lindquist et al. 2011).”*

References: Lindquist, K. A., Siegel, E. H., Quigley, K. S., and Barrett, L. F. (2013). The hundred-year emotion war: are emotions natural kinds or psychological constructions? Comment on lench, flores, and bench (2011). *Psychol. Bull.* 139, 255–263)
Jack, R., Garrod, O., and Schyns, P. (2014). Dynamic facial expressions of emotion transmit an evolving hierarchy of signals over time. *Curr. Biol.* 24, 187–192).

- The paragraph on page 6 lines 28 to 42 is hard to understand. A GA task was created? I though the authors would apply a genetic algorithm to expressions subjects provide? What does the no-constraints statement mean? Subjects are verbally prompted to express one out of four emotions aren't they? Isn't that pretty constrained (i.e. do what experimenter asks you to do)? How and which choices facilitate stimulus creation? I guess this whole paragraph needs rewriting. replacement, reconsideration.

We were not able to locate a comparable paragraph in the main text that matched up with the line and page references provided, but we have restructured the final paragraph of the introduction which mentioned 'experimenter constraints' to be clearer about the task procedure and the methods used. We have removed the term "constraints" and would like to clarify that we stated "our study developed a GA task" to indicate that the experimental procedure participants followed was the first time a GA has been applied in this way to psychological/emotional expression research (i.e. the cognitive task they were asked to complete did not exist before this study).

Page 5: *"To address this methodological gap, our study developed a cognitive task applying a GA to the individualised generation of facial emotional expressions. The task allowed participants to quickly and efficiently create facial expressions based on their own ideals and internal representation of a given emotion. Across a number of trials, participants were free to evolve their ideal emotional expressions according to their own perception, as opposed to having these ideals imposed by experimenter decisions according to pre-selected, standardised emotional stimuli. We applied this GA framework in which participants' choices directly facilitated stimulus creation, to quantify individual differences in emotion representation not typically captured by traditional paradigms where stimuli are pre-selected as part of the task design and are thus subject to experimenter-imposed bias."*

- The description of the procedure/method is not satisfactory at all. Critical aspects are provided elsewhere. The facial features are action units? The choices of features and how they change over iterations, the noise and mutations added all need to be described in an accessible way within this publication.

The reviewer states that "critical aspects are provided elsewhere", presumably referring to the technical methods that are described at length in [Reed, K., & Cosker, D. (2019). User-Guided Facial Animation through an Evolutionary Interface. *Computer Graphics Forum*, 38(6), 165–176] and cited throughout our paper. We have included a comprehensive description of these methods in the supplement of our paper, which aims to translate many of the technical aspects of the GA procedure that may not be accessible to a non-computer

graphics audience. However, we decided against including these details in full in the main text of the paper because (a) they are already available in published format in a paper that is referenced in our manuscript; (b), they would substantially increase the word count and detract from main methods and results presented, and (c) the methods are based in computer graphics and, we feel, could detract from our main message which concerns emotion recognition. Ultimately, the choice to include technical details in the supplement was so that the main text could be more accessible to a wider and non-specialist audience, as encouraged by Royal Society Open Science. To address the comment that the description of the methods was not satisfactory in the main text, we have incorporated the following information into the methods section regarding choice of blendshapes, mutation rate, and the coarse-to-fine approach employed by the GA. We think that this level of detail in the main text strikes the best balance between providing further key details, as the reviewer rightly suggests, whilst still keeping the main paper accessible to a broader audience who may not have a computational or psychophysics background. A more comprehensive outline of the methods is presented in the Supplement.

Page 7: “Blendshapes were chosen based on an animator’s (KR) expert opinion of the most significant shapes from a generic facial rig to cover a sufficient range of expression. The rig was adapted from a larger rig where blendshape correctives were removed and left/right blendshapes were combined into a single dual-sided action. This is necessary because without such constraints, correctives and asymmetry increase the likelihood of implausible, non-realistic expressions being created. To apply corrective shapes to improve realism of the faces generated, we sampled from rig controls, which incorporates correctives. This procedure is described at length in Reed & Cosker, 2019. Obtaining a suitable combination of blendshapes in the faces presented to participants also necessitates incorporating an appropriate mutation rate in the GA, described in terms of the difference in blendshape weights bounded $[0, 1]$. To achieve this, a random value was assigned between 0.25 and 0.75 (Reed & Cosker, 2019)(for complete details, see eAppendix-1).”

Page 7: “Participants completed six iterations of the GA, selecting 3 out of 10 faces on each iteration, with participants’ selections driving the feature selection scheme, as participants choose sample faces with the most desired characteristics for their search. The combination of six generations and ten faces per generation was chosen to strike a balance between avoiding information overload for participants on each trial, increasing efficiency, presenting an adequate number of generations (Frowd & Hancock, 2008), minimising participant ‘burn-out’ (i.e. boredom) since the faces become more similar with each iteration, and keeping the faces easy to view on a single screen (Reed & Cosker, 2019). To compensate for this relatively small population of test faces, we employed a coarse-to-fine approach to more extensively cover the search space. This process involves choosing facial features with the greatest variability for random sampling and mutation in initial generations, with more nuanced changes in expressions chosen in later generations (for extended technical details of the coarse-to-fine approach and overall GA evolution procedure, see eAppendix-1).”

Regarding the comment about action units, we would like to clarify that the facial features (i.e., blendshapes) chosen for our model were based on an animator’s rig and were not action units themselves, but their choice (42 out of a larger selection) was governed by the FACS action units to provide a translation to this system that many psychological

researchers of facial emotion may be more familiar with. In reality, blendshapes are specific to an artist-rendered facial rig, a system that those outside of the field of computer graphics/facial animation would likely be unfamiliar with.

- Am I right in understanding the design stating that participants always only saw pictures of themselves (along with some machine generated modifications [according to action units selected by participants as essential for showing one emotion or another])?

We would like to clarify that participants were not shown pictures of themselves. All participants were shown faces modelled based on the same artist-rendered facial rig, and features of these faces were modified across trials according to the parameters of the genetic algorithm, which accounted for participants' selections of faces on previous trials. We apologise that this was not made clearer in the text and have amended the task description as follows:

Page 7: "All participants were shown computer-generated faces modelled on the same artist-rendered facial rig."

- The power/sample size analysis seems to be for a purpose at best tangential to the present research questions. This section should also appear elsewhere and it should be tailored to the present research questions, preferably as a priori power analysis.

Power calculations were computed a priori based on the sample size required for the test-retest analyses, which was one of our key metrics to assess the consistency of within-participant responses to investigate individual differences. We have amended this text in the paper to clarify how power was assessed-

Page 10: "105 adults between 18 and 61 years of age (29% male, mean age 27+8 years, 51% Caucasian, 42% Asian, 2% Black, 5% Other) were recruited via a community convenience sample and completed the GA task on one occasion. An a priori power calculation revealed that in order to detect medium effect sizes (Cohen's $f=0.25$) with 95% power, a minimum sample size of $N=55$ was required. Therefore, anticipating 10% drop-off, 60 participants from the original convenience sample repeated the task on two more occasions, once approximately 30 minutes later, and again after 30-45 days."

We have kept the power calculation in the methods section where participant recruitment and sample sizes are outlined as this seems to be common practice in many papers published in the RSOS, but we would consider moving this to another location if the editor thinks there is a more suitable location for this information.

- In section 4.1. the authors begin with detailing their procedure for computing the discrepancy between the preferred participant face for a specific emotion and some 42 dimensional face-space. Much more detail on this computation is required.

We have now included additional details on the multivariate dispersion tests described in section 4.1.

Page 11: *“Multivariate dispersion analysis (Anderson 2006) assessed variability of ‘community structure’, i.e., variability of individual prototype faces from the population norm within each emotion category; the spatial median (i.e., population norm) was computed within 42-dimensional ‘face-space’ (Valentine et al. 2015) corresponding with the 42 blendshapes, and the average distance (‘dissimilarity’) between each individual data point and its category’s spatial median - defined in the principal coordinate of face-space - was calculated using Euclidian distances from this median as the dissimilarity measure. Levene’s F-statistic was calculated to compare the average distance of individual data points to the population norm for each emotion category, and the significance of dissimilarity (p-value) was calculated by permuting least-absolute-deviation residuals and recalculating the F-statistic.”*

- From figure two it seems as if lip corner depressor is clearly the most distinctive feature in both fear and sadness, relative to other action units. May be this is decisive for the issues in distinguishing between sad and fear and I doubt this is what the authors (should have) wanted.

We agree that the finding of the same blendshape, lip corner depressor, having the strongest influence in the prototype representations of fear and sadness was somewhat surprising based on the general conceptualisation in the literature of what constitutes fearful expressions. As such, we think this is one of the key findings of the paper, presenting new conceptualisations of facial features important for the representation and recognition of fear, particularly in relation to sadness. When individuals were allowed to subjectively generate expressions of their *own* representations (rather than relying on representations depicted by pre-selected and categorised stimuli), it was found that similar facial features are important in forming both sad and fearful expressions, a finding that was similarly reflected when other people were asked to identify these expressions. Given that facial emotion recognition tasks traditionally impose a pre-selected expression on the participants (rather than allowing them to generate their own expression), we argue that our GA method presents new and compelling evidence that this overlap between representations of fear and sadness may be one reason for discrepancies in the literature and failed replications of many traditional emotion perception tasks and may have significant implications for psychological research, where it is thought that aberrant emotion perception may be related to specific personality traits or mental health symptoms. Moreover, it has been repeatedly shown in the literature (as described on page 22) that fear and sadness are the most difficult expressions to correctly recognise, and our results suggest that one reason for this is the previously unidentified contribution of certain facial features to these expressions which overlap among categories.

We have added the following text to the discussion to further highlight this finding:

Page 22: *“Overlap and poorer identification for sad and fear is likely to reflect the fact that these expressions are inherently more ambiguous and complex (Goeleven et al., 2006), more*

context-dependent than happiness and anger (Barrett, Mesquita, and Gendron 2011), and are more likely to co-occur in daily life. Previous work has also shown that both sad and fearful expressions are associated with the action unit of stretched mouth (Kohler et al. 2004), which parallels our findings of lip corner depressor being a key blendshape for sad and fearful faces.”

- It wasn't easy to decode the design and it was harder still to try to make sense out of the decisions made by the authors. The text might have a much better flow, if the research question were made absolutely explicit. This could be for example the idiosyncratic nature of emotion expression across individuals. It could also be the use of GA to help participants approximate their personal emotion expression space. In the current version of the text both seem to be the case (and obviously there are a number of additional research questions the authors could pursue).

We agree that the research question(s) could be made clearer in the introduction. The reviewer is correct in both of their assertions: First, the study sought to present the use of a GA to allow participants to generate their own personal representations of emotional expressions for use in psychological research, as this has not been done before in the literature. However, this aim required the collection of task data for validation which allowed us to answer a second research question, which was whether this task reliably identified individual differences in emotion representation across the four categories. We have added the following text to the introduction to explicitly state these two research questions:

Page 6: *“This study had two primary objectives, 1) to present for the first time the use of a GA in facilitating participant-driven generation of facial emotional expressions, and 2) to use this task to examine individual differences in the way participants represent and perceive these expressions.”*

- The purpose of the external validity sample seems unclear to me. Something that is quite obscure at this stage is arguably validated with something that is even more obscure. The data also seem to be insufficiently cleared from nonsense observations.

We do not have any a priori reason to believe that participants' responses on the external validation study were a result of nonsense observations. This study followed a very similar format to many traditional tasks of emotion recognition in the literature, where subjects are shown a face and asked to identify the emotion being expressed. Similar to existing studies, we could have followed a more constrained format where participants were given e.g. 4 response options to choose from. This approach would perhaps ensure higher exact identification accuracy of each expression, but our results showed that even when response options are not provided and subjects are free to respond in any way they think is appropriate, expressions created by the GA procedure were largely validated, as confirmed by responses included in the respective word clouds for each emotion (figure 4A). In the context of this task, a “nonsense observation” would be a label that did not correspond to an emotional state/trait (termed “anomalous” responses in the paper). The number of

anomalous responses we recorded was exceedingly low (1% of total responses), despite the fact there was no forced-choice or experimenter expectation.

In the case of fear vs. sadness, there was less accuracy and consistency in the identification of these expressions, but these results mirror our findings from the initial study in terms of overlap in the individual prototype expressions that were created and likely would have been observed even in the context of a forced-choice version of this task (as fear and sad would both have been response options). Ultimately, the external validation provides further evidence for individual differences in facial emotion processing, showing that there is also population-level variability in emotion recognition as well as emotion representation.

- I am afraid this paper will need substantial revision and may be additional data. I like the gist of using GA for altering, modifying facial stimuli. Machine-based face generation might also be used to help subjects approximate their personal expression space. If this were part of the research questions, adequate display/visualization of such idiosyncrasies is essential.

We have now clarified in the methods section that our procedure did indeed use computer-based face generation to help subjects approximate their personal expression space (i.e. they were not shown photographs of their own faces):

Page 7: "All participants were shown computer-generated faces modelled on the same artist-rendered facial rig."

We think that the schematic presented in Figure 1 of the main text and the accompanying revised description of the rig and computer graphics procedure outlined in the methods also helps to clarify this and presents a comprehensive visualisation. We also present a visualisation of the expression approximations (i.e. prototype faces) generated by participants in Figure 2D of the main paper, and supplementary eFigure 2.

As reflected in our responses above, we have made substantial amendments and additions to the text that we hope clarify the aims and methods employed in our study. Thank you for the opportunity to revise and improve our paper.

Reviewer: 2

Comments to the Author(s)

The paper presents an interesting idea, but it raises some concerns in its present form that the authors should address, particularly the following.

The use of Genetic Algorithms (GAs) can provide a promising approach to study individual differences in (facial) emotion generation and perception. However, the use of interactive GAs to generate "personalized" emotional facial expressions is not new, and studies can be found for example in the fields of artificial life, artificial intelligence and animation since the mid-90s, which range in the types of faces and expressions used from the highly streamline drawings to morphing technology to 3D realistic animations. Moreover, recent work has used generative adversarial networks to generate contextually-relevant facial expressions in

dyadic human interactions. The authors should discuss such related work, both in terms of the interactive generation of individualized expressions and of the GAs / algorithms used.

We agree that the use of GAs to generate facial expressions is not new. We discuss the use of similar methods in e.g. computer science and animation on page 4:

“Participant-generated facial expressions have been of interest to the computer animation community for some time (Ersotelos and Dong 2008). Since the introduction of performance-driven animation (Williams 1990), there have been numerous implementations of this method to create realistic, performance (i.e. participant) driven graphics (Shin and Lee 2009; Yu, Garrod, and Schyns 2012)...New implementations of existing methods such as the Genetic Algorithm (GA)—thus termed because of its procedural similarities to evolutionary processes—may solve the longstanding problem of optimally yet efficiently capturing an individual’s internal representation of facial emotion...GAs have been used in ecology and forensic settings, where they enable eyewitnesses to generate facial composites with greater accuracy compared to traditional sketches (Frowd, Hancock, and Carson 2004; Gibson, Solomon, and Bejarano 2003). However, GAs have not been widely applied to facial emotion research.”

As highlighted in the final sentence of this text, the novelty in our approach centres on the fact that GAs have not been widely used in the psychological research domain of facial emotion processing, which has instead relied largely on standardised stimulus sets of facial expressions, thus overlooking participants’ own internalised representation of emotions. Indeed, an exciting feature of the present study is the novel application of this method to probe individual differences in socio-affective processing, with future implications for relating these measures to individual differences in e.g. personality measures or mental health symptoms. In our study, a GA is used for the first time to compare performance (i.e., expression generation) within a large population-based sample (rather than just assessing the GA output from a single individual) to study individual differences.

We have added the following text to this section to further highlight recent work that has been done using other methods for facial expression generation (also cited in our response to reviewer 1):

Page 4: *“Participant-generated facial expressions have been of interest to the computer animation community for some time (Ersotelos and Dong 2008). Since the introduction of performance-driven animation (Williams 1990), there have been numerous implementations of data-driven methods, including principle component analysis (PCA)(Blanz and Vetter 1999; Chai and Hodgins 2005), Gaussian Process Latent Variable Models (GPLVMs)(Grochow et al. 2004; Wang, Fleet, and Hertzmann 2007), and autoencoders (Fragkiadaki et al. 2015; Holden, Saito, and Komura 2016) to create realistic, performance (i.e. participant) driven graphics (Shin and Lee 2009; Yu, Garrod, and Schyns 2012). However, these techniques often rely on the secondary judgement of a skilled artist and are limited in their ability to allow users to explore and interpret resulting graphics. More recently, machine learning techniques such as Generative Adversarial Networks (GANs)(Goodfellow et al. 2014; Khan et al. 2021) and verbal crowd-shaping (Streuber et al. 2016) have been applied to create computer-generated facial expressions and body images based on sampled parameters and*

descriptive labels from a population. Whilst promising, these methods rely on unsupervised learning and classification, requiring vast training datasets and fully trained network for optimal performance. Therefore, such approaches may not be best-suited to index the nuanced and subjective nature of faces and emotion representation. Therefore, new implementations of existing methods such as the Genetic Algorithm (GA)...”

More details of the GA used and justification of the choices made should be discussed in the body of the paper, since the choice of the GA and its operators can have a big impact on the results obtained.

In line with another comment from Reviewer 1, we have now moved some of the key details from the Supplement to the main methods of the paper (the below text is also quoted in response to a comment from reviewer 1):

Page 7: “Blendshapes were chosen based on an animator’s (KR) expert opinion of the most significant shapes from a generic facial rig to cover a sufficient range of expression. The rig was adapted from a larger rig where blendshape correctives were removed and left/right blendshapes were combined into a single dual-sided action. This is necessary because without such constraints, correctives and asymmetry increase the likelihood of implausible, non-realistic expressions being created. To apply corrective shapes to improve realism of the faces generated, we sampled from rig controls, which incorporates correctives. This procedure is described at length in Reed & Cosker, 2019. Obtaining a suitable combination of blendshapes in the faces presented to participants also necessitates incorporating an appropriate mutation rate in the GA, described in terms of the difference in blendshape weights bounded [0, 1]. To achieve this, a random value was assigned between 0.25 and 0.75 (Reed & Cosker, 2019)(for complete details, see eAppendix-1).”

Page 7: “Participants completed six iterations of the GA, selecting 3 out of 10 faces on each iteration, with participants’ selections driving the feature selection scheme, as participants choose sample faces with the most desired characteristics for their search. The combination of six generations and ten faces per generation was chosen to strike a balance between avoiding information overload for participants on each trial, increasing efficiency, presenting an adequate number of generations (Frowd & Hancock, 2008), minimising participant ‘burn-out’ (i.e. boredom) since the faces become more similar with each iteration, and keeping the faces easy to view on a single screen (Reed & Cosker, 2019). To compensate for this relatively small population of test faces, we employed a coarse-to-fine approach to more extensively cover the search space. This process involves choosing facial features with the greatest variability for random sampling and mutation in initial generations, with more nuanced changes in expressions chosen in later generations (for extended technical details of the coarse-to-fine approach and overall GA evolution procedure, see eAppendix-1).”

However, my main concerns related to the fact that it is not clear which is the point the authors are trying to make and how the study addresses it. First, the authors criticize the use of "categorical" models and studies, as they overlook individual differences; however, the present study uses a categorical approach, which seems contradictory. Would the

authors be criticizing the use of categories or the universality claim? In any case, categorical/universal approaches do take into account individual differences in emotion expression and perception, and more recent studies (including by Ekman) can be found comparing "normative" aspects and individual differences (see for example recent publications by Ekman). Second, it is not clear what the contribution of this study to the study of individual differences is; large parts of the discussion focus on differences across expressions (categories) rather than trying to shed some light on individual differences, their context, and their significance. Finally, much like the "decontextualized" studies that the authors criticize, this study overlooks the role of the context in the expression, perception and interpretation of emotional expression, which is one of the key elements stressed by authors like Barrett-Feldman, which the authors quote to support the idea of the importance of individual differences.

We have now made a number of changes to the text to improve clarity of our research questions, study design, and interpretations of findings. We outline these changes below with regard to each of the reviewer's three points:

Regarding the first point about categorical approaches: we would like to clarify that our paper does not aim to criticise categorical approaches because, as the reviewer highlights, we follow these very approaches in this paper. Instead, we are trying to make the point that traditional studies of facial emotion perception, which are similarly based on categorical expressions, overlook important individual differences through the use of standardised stimulus sets which often impose experimenter bias. The reviewer's comment highlighted to us that we had not outlined this sufficiently clearly and we have now amended the text in the introduction as follows to clarify this point:

Page 4: "Methods for testing emotion recognition typically rely on the assumption that emotions are discreet categories, which requires broad agreement across observers. However, it is likely that emotional expressions are represented dimensionally within these categories (Calvo and Nummenmaa 2016), varying subtly from person to person. Many experiments are limited in their ability to index this variation within categories because they use pre-established sets of facial stimuli designed to universally represent basic emotions (Ekman 1982; Tottenham et al. 2009) when in reality, emotion categories are not represented in the same way by all individuals. Additionally, studies of emotion processing which use a forced-choice response to identify a given facial stimulus may artificially constrain the detection of individual differences (Gendron et al. 2014b). Therefore, the present study sought to understand this fine-grained variation in emotion representation at the individual level within traditional emotion categories through the indexing of participant-generated emotional facial expressions."

Regarding the second point about the study's contribution to the study of individual differences: we have moved the methods and results of analyses regarding within-subject response stability, which contributes to the investigation of individual differences within a population, from the supplement to the main text:

Page 12: "To assess reliability of the GA task, we examined whether between-subject individual differences in prototype faces were greater or smaller than within-subject

variance across time. This also allowed us to confirm that after six generations, the prototype face was not a result of either a specific mutation or local minimum in the algorithm's iterative process. We first calculated a dissimilarity matrix for each emotion from the Euclidean distances in 42-dimensional face-space between all the faces produced at the three time points. We then coded an individual identity matrix for each emotion, where each row and column were different faces, and within-subject comparisons were assigned a value of 0 and between-subject comparisons were assigned a value of 1. Mantel tests were used to test for correlations between the individual identity matrices and the dissimilarity matrices using the 'vegan' package in R (Oksanen et al., 2019) with 9999 permutations and using a Spearman test statistic. Resulting distance values from multivariate dispersion analysis within each emotion category (section 4.1) for each time point were also used to compute two-way mixed effects intraclass correlation coefficients (ICCs)."

Page 17: *"Mantel tests (Figure 3A) showed that faces generated for all four emotions were positively correlated with the individual-identity matrix [happy: $\rho = .048$, $p < .001$; sad: $\rho = .018$, $p < .001$; angry: $\rho = .010$, $p < .01$; fear: $\rho = .016$, $p < .001$], indicating that variation between subjects was greater than within-subject variability across the three time points. Moreover, there was no significant difference of correlations in each participant between time point 1 and time point 2, time point 2 and time point 3, or time point 1 and time point 3 for any emotion, indicating that the amount of time between visits (10 minutes vs. 4-6 weeks) did not significantly impact within-subject variability.*

ICCs calculated across the three time points using distances from the spatial median indicated moderate to good response reliability for all four emotion categories: happy: $ICC = .585$, $95\% CI = .363-.739$, $F(59, 118)=2.410$, $p < .001$; sad: $ICC=.576$, $95\% CI=.350-.733$, $F(59,118)=2.361$, $p < .001$; angry: $ICC=.419$, $95\% CI=.107-.634$, $F(59,118)=1.720$, $p < .01$; fear: $ICC=.574$, $95\% CI = .345-.731$, $F(59,118)=2.345$, $p < .001$ (Figure 3B)."

We think that this is a key strength of this study in terms of contributing to individual differences research, as it shows that responses on the GA task are stable across time and are able to distinguish among individuals within the population. We now highlight this point in the discussion:

Page 21: *"This characteristic further underscores the utility of this method for the investigation of individual differences, presenting scope to distinguish between individuals within a population."*

Regarding the final point about the role of context: we agree that role of context in the representation of expressions is a very important consideration, and we have added the following text to the discussion to highlight this:

Page 24: *"We also acknowledge the importance of context in the representation, perception and interpretation of emotional expressions (Barrett, Mesquita, and Gendron 2011; Dolan 2002), despite the fact that much emotion research operates on the assumption that basic emotion categories are universally recognised and perceived as intended by the experimenter. Our current task partially addresses this limitation by allowing subjects to generate representations of their own ideals of facial emotions, but future studies should*

examine these representations across various stimulus-based, perceiver-based, and cultural contexts which likely impact the encoding and perception of facial expressions.”